# Positive Effects of a Mediterranean Diet Supplemented with Almonds on Female Adipose Tissue Biology in Severe Obesity

**DOI:** 10.3390/nu14132617

**Published:** 2022-06-24

**Authors:** Óscar Osorio-Conles, Romina Olbeyra, Violeta Moizé, Ainitze Ibarzabal, Oriol Giró, Judith Viaplana, Amanda Jiménez, Josep Vidal, Ana de Hollanda

**Affiliations:** 1Centro de Investigación Biomédica en Red de Diabetes y Enfermedades Metabólicas Asociadas (CIBERDEM), Instituto de Salud Carlos III (ISCIII), 28029 Madrid, Spain; oosorio@clinic.cat (Ó.O.-C.); vmoize@clinic.cat (V.M.); viaplana@clinic.cat (J.V.); 2Institut d’Investigacions Biomèdiques August Pi i Sunyer (IDIBAPS), 08036 Barcelona, Spain; rolbeyra@clinic.cat (R.O.); giroi@clinic.cat (O.G.); ajimene1@clinic.cat (A.J.); 3Obesity Unit, Endocrinology and Nutrition Department, Hospital Clínic de Barcelona, 08036 Barcelona, Spain; 4Gastrointestinal Surgery Department, Hospital Clínic de Barcelona, 08036 Barcelona, Spain; aibarza@clinic.cat; 5Centro de Investigación Biomédica en Red de la Fisiopatología de la Obesidad y Nutrición (CIBEROBN), Instituto de Salud Carlos III (ISCIII), 28029 Madrid, Spain

**Keywords:** Mediterranean diet, almonds, obesity, adipose tissue, inflammation

## Abstract

It has been suggested that weight-loss-independent Mediterranean diet benefits on cardiometabolic health and diabetes prevention may be mediated, at least in part, through the modulation of white adipose tissue (WAT) biology. This study aimed to evaluate the short-term effects of a dietary intervention based on the Mediterranean diet supplemented with almonds (MDSA) on the main features of obesity-associated WAT dysfunction. A total of 38 women with obesity were randomly assigned to a 3-month intervention with MDSA versus continuation of their usual dietary pattern. Subcutaneous (SAT) and visceral adipose tissue (VAT) biopsies were obtained before and after the dietary intervention, and at the end of the study period, respectively. MDSA favored the abundance of small adipocytes in WAT. In SAT, the expression of angiogenesis genes increased after MDSA intervention. In VAT, the expression of genes implicated in adipogenesis, angiogenesis, autophagy and fatty acid usage was upregulated. In addition, a higher immunofluorescence staining for PPARG, CD31+ cells and M2-like macrophages and increased ADRB1 and UCP2 protein contents were found compared to controls. Changes in WAT correlated with a significant reduction in circulating inflammatory markers and LDL-cholesterol levels. These results support a protective effect of a Mediterranean diet supplemented with almonds on obesity-related WAT dysfunction.

## 1. Introduction

The Mediterranean diet (MD) has been associated with decreased cardiovascular disease (CVD) risk, reduced type 2 diabetes incidence, improvements in lipid profile and blood pressure, and reduced levels of systemic inflammation, changes that occur even in the absence of significant weight loss [1]. Thus, higher adherence to the traditional MD is associated with transitions to healthier obese phenotypes [2] and inversely related to long-term complications of diabetes [3] and several cardiovascular risk factors [4]. It has been proposed that these findings may shift the paradigm of CVD prevention, with changes in diet composition becoming an important component as, in clinical practice, switching to MD can be more attainable than pursuing weight loss [5,6,7]. In addition, MD supplemented with tree nuts, which are important components of the MD [8], has demonstrated a better performance in lowering blood atherogenicity [9], the prevalence of metabolic syndrome [10] and the cumulative incidence of stroke versus MD supplemented with extra-virgin olive oil [11]. Such positive effects of MD supplemented with nuts on CVD risk factors were found after only 3 months of follow-up [12]. 

Almonds are rich in monounsaturated fat, fiber and polyphenols. Almond polyphenols, mainly composed of tannins and flavonoid [13], are bioavailable and extensively biotransformed by the microbiota and host tissue upon consumption [14] and have well-documented activity in reducing inflammation and oxidative stress [15]. Among Mediterranean tree nuts, almonds are especially rich in α-tocopherol [8,16], a fat-soluble antioxidant vitamin with known protective effects on obesity, metabolic syndrome and lipid levels [17]. However, although the health benefits of MD and almonds are well-proven, the precise mechanisms mediating these effects are incompletely understood. 

White adipose tissue (WAT) dysfunction has been proposed as an underlying factor of the metabolic disturbances associated with obesity. The obesity-related WAT dysfunction encompasses an altered adipokine secretome [18], unresolved inflammation [19], dysregulated autophagy [20,21], inappropriate extracellular matrix remodeling and insufficient angiogenic potential [19]. It has been suggested that a larger proportion of hypertrophic adipocytes in WAT favors unhealthy WAT tissue expansion, due to persistent hypoxia associated with limited oxygen diffusion in the context of larger adipocytes [19]. A higher abundance of infiltrating macrophages forming crown-like structures around single adipocytes has also been described in WAT from patients with obesity [20]. These macrophages predominantly present the M1 pro-inflammatory phenotype and promote inflammation by releasing granulocyte-macrophage colony-stimulating factor (GM-CSF), tumor necrosis factor alpha (TNF-α), IL-1β or IL-6, contributing to insulin resistance. On the other hand, activated M2-like macrophages play a role in WAT expansion, thermoregulation, antigen presentation and iron homeostasis, secreting anti-inflammatory cytokines [21].

Finally, beiging is the process through which WAT can change its phenotype to a brown-like adipose tissue known as beige/brite adipose tissue. Previous studies have focused on the ability of specific dietary components, such as polyphenols, in enhancing energy expenditure by activating brown adipose tissue or promoting WAT beiging [22,23]. Nevertheless, the potential contribution of an MD pattern in this regard has, to our knowledge, never been evaluated.

The aims of our study were to investigate the short-term effects of an MD-based intervention supplemented with almonds (MDSA) on the main features of WAT dysfunction and the association between tissue variables of WAT dysfunction and systemic markers of metabolic health in women with obesity that were bariatric surgery (BS) candidates.

## 2. Materials and Methods

### 2.1. Study Design and Subjects

This randomized, two-arm, single-center, exploratory study was conducted at a tertiary University Hospital. It was approved by the Institutional Ethics Committee, and written consent was obtained from all study participants. Women candidates for BS, aged 18–68 years, with a BMI of 40–50 kg/m^2^ and two additional features of metabolic syndrome (fasting glucose 100–126 mg/dL, arterial blood pressure >130/85 mmHg, HDL <50 mg/dL, triglycerides >150 mg/dL) and stable weight during the 3 previous months were consecutively invited to participate in the study. Exclusion criteria were tree nut allergy, type 2 diabetes, treatment with metformin or corticosteroids (except inhaled corticosteroids), regardless of indication, active pharmacological treatment related to weight gain or weight loss, previous BS, abnormal thyroid function and lack of commitment to follow the protocol. A total of 38 women candidates to BS were 1:1 assigned by simple randomization to the MDSA or control group according to a computer-generated list of sequential random allocation. A call to the coordinator office ensured that the treatment was assigned correctly according to the randomization list. The investigators that performed the histological and analytical studies were blinded to the allocation of intervention groups by using successive patients’ codes. Given the absence of previous studies evaluating the role of MDSA on WAT dysfunction, it was not possible to formally calculate the sample size; therefore, this study is considered exploratory. The recruitment period was from July 2018 to December 2019. Two participants in the MDSA group were receiving statin treatment. From the control group, 3 were on statin and 1 on ezetimibe therapy.

### 2.2. Nutritional Intervention

After a 3-month weight stabilization phase during which maintenance of the participant’s usual dietary pattern was encouraged, participants were randomly assigned to the almond-supplemented MD group (MDSA) or maintenance of usual diet (control) group in which no changes in dietary habits were advised. All participants were followed by a nutritionist every 2 weeks during the 3-month stabilization phase and the 3-month intervention phase. Total caloric intake was estimated and adjusted to ensure body weight was stable throughout the stabilization and intervention phases of the study. Physical activity was not promoted. Following randomization, participants in the MDSA group received raw unpeeled almonds at no cost for the entire study (equivalent to 30 g/d). The nutritional composition of almonds per 100 g was: 628 kcal; total fat—56 g; saturated fat—4.9 g; carbohydrate—2.2 g; sugars—2 g; dietary fiber—9.8 g; protein—24 g; salt—0 g; vitamin E—17 mg; calcium—223 mg; phosphorus—458 mg; magnesium—232 mg; iron—3 mg. Instructions were given about how to increase MD adherence in the MDSA group in order to increase the use of olive oil for cooking and dressing, consumption of fruit, vegetables, fish and white meat instead of red or processed meat and to promote the preparation of homemade sauce with tomato, aromatic herbs, onion, garlic, and olive oil to dress vegetables, pasta, rice, or other dishes. At study inclusion and every 2 weeks, dieticians delivered individual sessions consisting of informative talks and provided written material with elaborated descriptions of typical MD foods, seasonal shopping lists, meal plans and recipes. A previously validated 14-item Mediterranean Diet Adherence Screener (MEDAS) was used to assess adherence to MDSA at baseline and at months 1, 2 and 3 of the study [24]. α-Linolenic acid (ALA) relative content of red blood cell membranes was measured as a biomarker of nut consumption by gas chromatography using an Agilent 7890 A Gas Chromatograph (Agilent España, Spain), as previously described [25].

### 2.3. Examinations and Calculations

Anthropometric measures (body weight, waist circumference, and body mass index) and blood pressure were collected following standardized procedures at baseline and at the end of the dietary intervention period.

### 2.4. Mixed Meal Tolerance Test

Patients attended the research facility before and after the dietetical intervention. After an overnight fast, a cannula was inserted into the distal forearm; blood samples were withdrawn at baseline for glucose, insulin, lipidic profile and inflammatory circulating molecules measurement. Patients were then asked to ingest a 250 mL standard liquid mixed meal (SLMM; Isosource Energy, Novartis, Switzerland; contained 398 kcal, with 50% of calories being carbohydrates, 15% protein and 35% fat) over 5 min. Additional blood samples were obtained at 30, 60, 90 and 120 min after meal ingestion for insulin and glucose measurements. The following indexes were calculated with these data: HOMA-IR, Matsuda Index, Insulinogenic Index and Disposition Index [26].

### 2.5. Body Composition

Total body fat and lean mass were measured by dual-energy X-ray absorptiometry (DXA) using a GE Lunar iDXA with the software enCORE provided by the manufacturer (GE Healthcare, Madison, WI, USA). The software was also used to calculate estimated visceral fat (eVAT) in the android region from the following formula: total adipose fat mass in the android region = eVAT + estimated subcutaneous fat in the android region [27].

### 2.6. Adipose Tissue Biopsies

At baseline, a subcutaneous WAT (SAT) biopsy was performed following an 8 h fast. About 1 g of superficial adipose tissue was obtained, under local anesthesia with 1 mL mepivacaine 2%, through a 2 cm incision in the periumbilical zone under the edema area. At the end of the dietary intervention (<7 days after completion), participants underwent BS, and the same surgeon collected a second sample of SAT, from the same region and depth, and a sample of visceral WAT (VAT) from the distal portion of the omentum majus by surgical excision. Collection was performed before the specific bariatric procedure began. WAT samples were collected in DMEM and rinsed in PBS. A portion was immediately frozen before RNA analysis. The other part was fixed overnight at 4 °C in 4% paraformaldehyde and processed for standard paraffin embedding. Starting at the tissue apex, 3 sections of 3 μm thick were cut at a minimum of 100 μm intervals across the sample tissue. Serial sections were matched for additional independent analyses. All analyses on WAT and blood samples were performed by an investigator blinded to patient allocation.

### 2.7. Fat Cell Area and Adipose Tissue Fibrosis

Hematoxylin and eosin staining was conducted to assess adipocyte morphology. A minimum of 5 pictures were taken from each sample at 10× magnification under a Nikon Eclipse E600 microscope. The area (μm^2^) of at least 2000 cells was digitally analyzed using ImageJ and the “MRI Adipocytes Tools” toolset (http://dev.mri.cnrs.fr/projects/imagej-macros/wiki/Adipocytes_Tool, accessed on 7 June 2021). Sirius red staining was used for the quantification of pericellular fibrosis at 20× magnification in at least 10 images per sample. Automated analysis of the captured images was carried out using ImageJ and the “MRI Fibrosis Tool” (http://dev.mri.cnrs.fr/projects/imagej-macros/wiki/Fibrosis_Tool, accessed on 7 June 2021) and expressed as the ratio of fibrous tissue area stained with picrosirius red relative to the total tissue surface.

### 2.8. Gene Expression

Total RNA was isolated using RNeasy Lipid Tissue Mini Kit (Qiagen, Hilden, Germany). Concentration and purity were measured using a NanoDrop 1000 spectrophotometer (Thermo Scientific, Waltham, MA, USA). Equal amounts of RNA from SAT and VAT (2 μg) were reverse-transcribed using the Superscript III RT kit and random hexamer primers (Invitrogen, Carlsbad, CA, USA). Reverse transcription reaction was carried out for 90 min at 50 °C and an additional 10 min at 55 °C. Real-time quantitative PCR (qPCR) was performed with a 7900HT Fast Real-Time PCR System (Applied Biosystems, Foster City, CA, USA) using GoTaq^®^ qPCR Master Mix (Promega Biotech Ibérica, Madrid, Spain). Expression relative to the housekeeping gene RPL6 was calculated using the delta C_t_ (DC_t_) method. Gene expression is presented as the 2^(−DC_t_) values for VAT samples and as the log_2_ fold change values for SAT samples. The list of primers used in this study is provided in Appendix A.

### 2.9. Immunofluorescence

Immunofluorescence staining was performed in WAT preparations collected from eight subjects from each group according to the standard protocol using the following antibodies and dilutions: CD206/MRC1 (ab125028, Abcam, Cambridge, UK; rabbit, 1:200), PLIN1 (ab60269, Abcam, goat, 1:200), CD31 (sc-376764, Santa Cruz, Dallas, TX, USA; mouse, 1:200), VEGFA (ab46154, Abcam, rabbit, 1:200), UCP2 (MA5-31946, Thermo-Fisher, Waltham, MA, USA; mouse, 1:200), UCP3 (ab180643, Abcam, rabbit, 1:100), PPARG (ab45036, Abcam, rabbit, 1:200) and ADRB1 (PA1-049, Thermo-Fisher, rabbit, 1:100). Tissue sections were rehydrated and subjected to heat-mediated antigen retrieval in citrate buffer. After a blocking step in 5% donkey (#017-000-121) or goat serum (#005-000-121, Jackson Immunoresearch, West Grove, PA USA) and permeabilization using 1% (v/v) Triton X-100, tissue sections were incubated overnight with primary antibodies. Then, they were incubated for 1 h with appropriate secondary antibodies conjugated with Alexa Fluor 488 (#711-546-152, Jackson Immunoresearch, 1:400) or 555 (#A21435, Invitrogen, 1:400) and counterstained with Hoescht 33258 (Sigma-Aldrich, St Louis, MO, USA) for the staining of nuclei. The immunofluorescent images were visualized and captured using a Nikon Eclipse E600 Fluorescence Microscope, collected with Olympus Cell^D software v3.4 and subsequently analyzed using Image J v1.50d software (Wayne Rayband, National Institutes of Health, http://rsb.info.nih.gov/ij/, accessed on 7 June 2021).

### 2.10. Immunoblotting

VAT samples from twelve subjects from each group were lysed using RIPA lysis and extraction buffer and centrifuged at 18,000× *g*, 4 °C for 20 min. A total of 20 µg of protein was resolved by SDS-PAGE (10%) and then transferred to a Polyscreen PVDF membrane (Perkin Elmer, Waltham, MA, USA), blocked in 5% non-fat milk powder added to TRIS-buffered saline with 0.05% Tween 20 (TBST) and further incubated overnight at 4 °C with the primary antibodies cited above at 1:1000 dilution. Actin (A2066, Sigma-Aldrich; 1:1000) was used as loading control. Following a wash in TBST solution, membranes were incubated with horseradish peroxidase-conjugated anti-rabbit (NA934, GE Healthcare, Chicago, IL, USA) or anti-mouse (NA931, GE Healthcare) antibodies and visualized with Immobilon Forte western HRP substrate (Millipore, Burlington, MA, USA) using a LAS4000 Lumi-Imager (Fuji Photo Film, Valhalla, NY, USA). Protein spots were quantitated with Image J software.

### 2.11. Circulating Levels

Serum levels of GM-CSF, IFN-γ, IL-6, TNF-α, IL-1β, sE-Selectin, adiponectin, sICAM-1, sVCAM-1 and SAA were measured in plasma samples collected before and after the dietary intervention using magnetic bead Milliplex MAP™ custom panels (EMD Millipore, Burlington, MA, USA) following the supplier’s instructions. Data from the reactions were acquired using the Luminex 100™ System (Luminex, Austin, TX, USA) and analyzed as fluorescence intensity. Thereafter, data were processed and analyzed with the Milliplex Analyst™ v.5.1.0.0 standard, 2012 (Merck Millipore KGaA, Darmstadt, Germany) and presented as target concentrations. Intra- and inter-assay %CV for hematological and biochemical measurements are provided in Appendix A.

### 2.12. Statistical Analysis

Data are presented as mean ± SD or *n* (%) when appropriate. Normal distribution of variables was tested using the Shapiro–Wilk normality test. The effect of the nutritional intervention on anthropometric (weight, BMI, waist circumference, body fat%, estimated SAT and VAT) and metabolic variables (FPG, insulin, insulin sensitivity and secretion, and lipid profile) and circulating levels was assessed by Two-Way Repeated-Measures ANOVA. To assess differences between groups, Student’s *t*-test or Mann–Whitney U test was used as appropriate for gene (SAT fold changes and VAT mRNA levels) and protein expression. The Holm–Sidak method or Bonferroni post hoc were used to correct for multiple comparisons. All analyses were adjusted for baseline values except for VAT tissue variables and SAT immunofluorescence intensity values. As group allocation could be confounded by dietary adherence, analyses were performed both based on group allocation (MDSA versus control group) as well as according to MDSA adherence measured with MEDAS (highest versus lowest tercile) irrespective of group allocation. Spearman’s test and multiple linear regression adjusted for confounding factors (age, BMI) were performed to assess the association between MEDAS and outcome variables. All statistical tests were performed using SPSS version 25.0 software (IBM Corp., New York, NY, USA) and GraphPad PRISM 6.0. Statistical significance was defined as a *p*-value below 0.05.

## 3. Results

Mean age and BMI at baseline were 47.2 ± 11.3 years and 44.7 ± 3.6 Kg/m^2^, respectively. Two participants from the control group dropped out of the study due to lack of commitment to keep appointments. Groups were well matched for additional variables (Table 1). No significant changes in body weight were observed in either group throughout the intervention. Both groups showed comparable low scores in MEDAS at baseline (<7 points, Appendix A). Patients allocated to the MDSA group showed a statistically significant increase in MEDAS throughout the study, which was accompanied by a raised ALA relative content in RBC membranes at the end of the intervention (Appendix A). No side effects of the nutrition intervention or due to biopsy collection were observed during the study.

### 3.1. Adipose Tissue Morphology

Mean SAT- and VAT-adipocyte area did not significantly change after the intervention in either group (Appendix A). Nevertheless, analysis of the adipocyte size distribution revealed that MDSA favored the abundance of small adipocytes in SAT (Figure 1A,B). Similarly, the adipocyte size distribution in VAT showed the enrichment of smaller adipocytes in the MDSA group compared to controls (Figure 1C).

Sirius red staining on formalin-fixed sections showed that fibrosis around adipocytes (i.e., pericellular fibrosis) remained unchanged in both depots after the dietary intervention (Appendix A).

### 3.2. Changes in Gene Expression in SAT

A gene-expression analysis of genes involved in inflammation, adipogenesis, autophagy, fatty acid (FA) metabolism, FA oxidation (FAO), adipocyte britening, glucose metabolism and adipokines was performed in SAT. Intervention-associated changes were comparable in the MDSA and control groups (Appendix A). Among these genes, only the leptin receptor (*LEPR*, Figure 2A) and autophagy-related gene 5 (ATG5, Figure 2B) showed differential albeit not statistically significant expression patterns (*p* = 0.054 and *p* = 0.066, respectively). The relative mRNA expression of *TGFB1* did not change throughout the study (Appendix A). On the contrary, the expression of the angiogenesis-related genes *PDGFRB*, *VEGFA*, *VEGFR1* and *VEGFR2* was significantly increased after MDSA intervention compared to controls (Figure 2C). A higher VEGFA protein expression at the end of the study was confirmed by immunofluorescence staining (Figure 2D), while CD31 protein abundance was not significantly different among groups (*p* = 0.056, Figure 2E).

### 3.3. Differences in Gene Expression in VAT

A larger expression of the pan-macrophage marker CD68 and the M2-type macrophage markers MSR1/CD204 and MRC1/CD206 was found in VAT from subjects in the MDSA group (Figure 3A). Of note, M1-type macrophage marker CD80 expression was not different. These differences were confirmed by the quantitative analysis of MRC1 immunofluorescence staining in VAT sections (Figure 4A). Conversely, MDSA group allocation was not associated with differences in the mRNA expression of cytokines (Figure 3A).

As in SAT, the gene expression of *PDGFRB* was increased in VAT from women in the MDSA group. Larger *VEGFB* expression, but not *VEGFA* expression, was also found (Figure 3B). Additionally, a higher presence of CD31 positive cells was detected by immunofluorescence in tissue sections (Figure 4B). The mRNA gene expression of the adipogenesis marker *PPARG* was larger in the MDSA group (Figure 3C) and such an increase was confirmed at the protein level by immunoblotting (Figure 3H) and immunofluorescence staining (Figure 4C) of the PPARG2 isoform. The expression of senescence-related genes p16, p21 and p53 was comparable across groups. However, we found an increased expression of autophagy-related ATG7 and ATG12 in VAT from the MDSA group (Figure 3D), while ATG5 showed a non-significant trend (*p* = 0.054). MDSA also had no effect on the visceral expression of various glucose metabolism-related genes and adipokines (Appendix A).

Finally, we also screened several genes implicated in the different steps of FA metabolism. Interestingly, MDSA was associated with a larger expression of the lipolysis-regulating genes *PLIN1*, *ATGL* and *MGLL* (Figure 3E) and this was accompanied with a higher expression of β-adrenoceptor 1 (*ADRB1*) at mRNA (Figure 3G) and protein level, measured by Western blotting (Figure 3H) and immunofluorescence staining (Figure 5A). Similarly, genes related to FAO and thermogenesis (*PPARA*, *PGC1A*) and mitochondrial succinate dehydrogenase complex flavoprotein subunit A (*SDHA*) appeared to be positively modulated (Figure 3F). The white adipocyte britening marker *PRDM16* was upregulated (Figure 3G), while *CIDEA* showed a trend towards a larger expression (*p* = 0.07). Despite the *browning* marker gene UCP1 expression being comparable, an increased expression of the uncoupling proteins UCP2 and UCP3 after MDSA was observed. Larger protein levels of UCP2 were confirmed by immunoblot analysis (Figure 3H) and immunofluorescence (Figure 5B), while UCP3 content was undetectable by Western blot and showed a non-significant trend after immunofluorescence staining of VAT sections (Figure 5C). Furthermore, the MDSA group showed larger mRNA expression of *ABCA1* and *APOE* genes, implicated in cholesterol efflux and HDL formation (Figure 3E).

### 3.4. Clinical and Circulating Parameters

At the end of the three-month intervention, anthropometrical parameters, body composition and insulin resistance indexes were unchanged. Still, total cholesterol and LDL cholesterol were reduced in the MDSA group, as well as several inflammatory parameters such as GM-CSF, IFN-γ and IL-1ß. Soluble cell adhesion molecules (sCAMs), adiponectin and other inflammatory markers remained unchanged in both groups (Table 1).

### 3.5. Associations with Mediterranean Diet Adherence Screener (MEDAS)

As mentioned above, we also performed an exploratory analysis of data based on MDSA adherence irrespective of group allocation. MD adherence was estimated from the MEDAS (highest tertile, ≥11 points, versus lowest tertile, ≤6 points). This analysis was concordant with the metabolic and inflammatory findings mentioned above. In the subgroup with the highest MEDAS punctuation, intervention resulted in a significant reduction in IL6, IL1ß, INF-γ, GM-CSF, and TNFα levels, compared to subjects with the lowest MEDAS punctuation (all *p* < 0.05, data not shown). Additionally, and similarly to the allocation-group comparison, no differences were found in adiponectin, sICAM-1, sVCAM-1 and SAA according to MD adherence.

Finally, positive associations between MEDAS and VAT expression of *ABCA1*, *PPARA*, *PGC1A*, *ADRB1* and *ADRB3* were found (Table 2).

## 4. Discussion

In our study, a short-term MDSA intervention was associated with changes in female WAT biology, especially in the visceral depot. Most of these effects revolve around the amelioration of characteristic features of obesity-induced WAT dysfunction and were accompanied by increased anti-inflammatory macrophage infiltration in VAT. The changes in adipose tissue biology occurred concomitantly with clinical changes, namely, an improvement in lipidic profile and decreased inflammatory circulating molecules. Despite the short term, MDSA did not modify insulin resistance indexes in our study, changes in other metabolic parameters and, perhaps even more significant, in WAT morphology occurred within three months after the adoption of an MDSA.

Presence of hypertrophic, insulin-resistant adipocytes is one of the main features of WAT in the context of obesity. After an MDSA intervention, both WAT depots showed a shift towards a smaller adipocyte population, despite no changes in body composition estimated by DXA were found, reflecting fat redistribution into newly recruited preadipocytes. This phenomenon was accompanied by an upregulation of *PPARG*, *PPARA* and lipolysis-related genes in VAT. *PPARG* is a master regulator of adipocyte biology and modulates lipid metabolism through the release, transport and storage of free FAs [28]. *PPARG2* is the splice variant with the highest adipogenic activity and is exclusively localized in WAT [29]. Likewise, *PPARA* is important for inducing adipocyte mitochondrial biogenesis, upregulating genes involved in FAO and limiting proinflammatory signaling during chronic lipolytic activation [30]. Both PPARG and PPARA agonisms promote the white-to-brite conversion in human adipocytes [31].

We also described an upregulation of *ADRB1* in VAT after MDSA. It is known that various subtypes of β-adrenergic receptors are present in WAT; their agonism leads to the proliferation of brown adipocytes (*ADRB1*) [32] and lipolysis and/or of thermogenesis activation (*ADRB2* and *ADRB3*) [33]. Interestingly, recent studies state that *ADRB1*, and not *ADRB3*, might be the primary regulator of brown adipocyte metabolism in humans [34]. WAT *beiging* has been associated with large-scale tissue remodeling, including increased micro-capillary formation, nerve innervation and modulation of immune cell populations [35]. Despite the fact that the expression of UCP1, the specific marker of brown and beige adipocytes, remained unaltered in our study, other thermogenic markers such as *PGC1A* and *PRDM16* increased after MDSA in VAT. Of note, several UCP-1 independent thermogenic mechanisms have been identified in recent years [36]. *PGC1A*, the master regulator of mitochondrial division [37], controls thermogenic gene activation in response to cold and β-adrenergic agonists [38] and interacts with *PPARA* to coactivate target genes involved in mitochondrial FAO [39]. On the other hand, *PRDM16* specifically regulates the induction of brown-fat-specific genes during the differentiation process by binding and enhancing the transcriptional function of *PPARG*, *PPARA* and *PGC1A* [40,41].

Unlike UCP1, the uncoupling protein UCP2 was upregulated after MDSA while the increase in UCP3 was only significant at the mRNA level. Recent findings involve UCP2 in the transport of four-carbon mitochondrial substrates (e.g., malate, oxaloacetate, and aspartate) outside the mitochondria [42], which allows the regulation of mitochondrial substrate oxidation and reactive oxygen species (ROS) levels regardless of mitochondrial uncoupling [43,44]. On the other hand, several studies reported an important role of UCP3 in FAO and oxidative damage prevention by mitochondrial ROS [45,46,47], stating that UCP3 may facilitate FAO by transporting FAs into the mitochondria [48], thus proposing a protective role for this protein in obesity [49]. Of note, both UCP2 and UCP3 activation efficiency increases with increased FA unsaturation and chain length [50], features that FAs predominantly found in MD fulfill.

In our study, increased lipolysis was accompanied by an upregulation of *ABCA1* and apolipoprotein E (*APOE*) genes, which play a major role in HDL biogenesis and thereby are candidate mediators of MD-related atheroprotective effects. ABCA1 is an integral cell-membrane protein that mediates the rate-limiting step of high-density lipoprotein (HDL) biogenesis and suppression of inflammation by triggering signaling pathways through interaction with an apolipoprotein acceptor [51]. APOE has long been known to be atheroprotective, mainly because of its ability to promote the removal of atherogenic lipoproteins from the circulation and the formation of APOE-containing HDL particles [52]. Interestingly, lower LDL cholesterol circulating levels were found after MDSA.

MDSA was associated with the VAT expression of autophagy genes *ATG5* and *ATG7*. A protective role has been proposed for autophagy in the context of obesity [53,54,55]. It has been speculated that some of the beneficial effects of MD could be partly explained by the ability to activate the autophagy of some of its components [56]. Thereby, polyphenols present in some of the main ingredients of the MD were shown to enhance autophagy, such as resveratrol, present in grapes, wine and some nuts [57,58], and oleuropein or oleocanthal, present in extra-virgin olive oil [59]. Almonds, in particular, among Mediterranean tree nuts, are especially rich in α-tocopherol [8,16], another well-known antioxidant. Previous clinical studies have verified the modulatory effects of almonds on serum glucose, lipid levels, the regulatory role on body weight, and protective effects against diabetes, obesity, metabolic syndrome and CVD [15,16,60].

Despite the short duration of the intervention, these changes in WAT were accompanied by a significant decrease in systemic inflammation biomarkers and improvements in lipid profile. In one of the first MD studies, Estruch et al. showed a decrease in circulating IL6, C-reactive protein and total cholesterol after a 3-month intervention, while HDL cholesterol levels raised [12]. In accordance with previous reports by other authors, we found reduced levels of total cholesterol [12,61,62], LDL cholesterol [61,62], IFN-γ, GM-CSF and IL-1ß [63] after MDSA. Although previously described as downmodulated by MD [12], circulating sICAM-1 remained unaltered in our study, albeit its levels were negatively correlated with MEDAS. Nevertheless, we could not demonstrate any effect of a 3-month MDSA intervention on markers of insulin resistance or body composition. This could be related to the short intervention duration and the relative mild insulin resistance at baseline. Additionally, the small number of subjects could explain the absence of changes in these two clinical parameters.

We acknowledge this study is not without limitations. As mentioned above, the limited number of subjects included in our exploratory study could have hampered our ability to find significant changes in several of the parameters included in the analysis. However, to the best of our knowledge, this is the first study that comprehensively addresses the study of adipose tissue function following an MD intervention. Second, it could be argued that our dietary intervention was too short to result in significant modifications of clinical parameters. This means that such a short-term study does not necessarily imply that the findings persist for years. However, the results of the present study are strongly indicative of a beneficial effect of the MDSA also on a long-term basis. Third, our study population was limited to women with severe obesity. We chose to include women only to avoid the confounding effect of gender in the context of a study with a small sample size and this limits the scope for drawing conclusions on a male population. Of note, differences between groups were more evident in VAT and this adipose tissue compartment is only available in subjects scheduled for abdominal surgeries, as BS. Obviously, access to VAT only at the end of the intervention could be viewed as an additional limitation. However, we deem it unlikely that major differences prior to intervention explain our findings as study groups were well matched for body composition and laboratory data.

## 5. Conclusions

In summary, a short-term MD intervention supplemented with almonds was associated with an increased expression of adipogenesis, angiogenesis and autophagy-related genes in VAT compared to controls. This was accompanied by an upregulation of FA utilization and several markers of thermogenesis. Some of these changes point to a diminished ROS production and enhanced tissue health in the VAT depot. The appearance of a small adipocyte population and the higher presence of M2-like anti-inflammatory macrophages may be explained in part by an MDSA-dependent modulation of adipocyte metabolism. These changes in WAT biology were accompanied by systemic benefits, such as decreased inflammatory circulating levels and decreased total and LDL-cholesterol levels.

Overall, our results support a protective effect of the MD supplemented with almonds in obesity-related metabolic complications and provide potential mechanistic links between MD-induced improvements on WAT biology and the proven systemic benefits of this dietary pattern.

## Figures and Tables

**Figure 1 nutrients-14-02617-f001:**
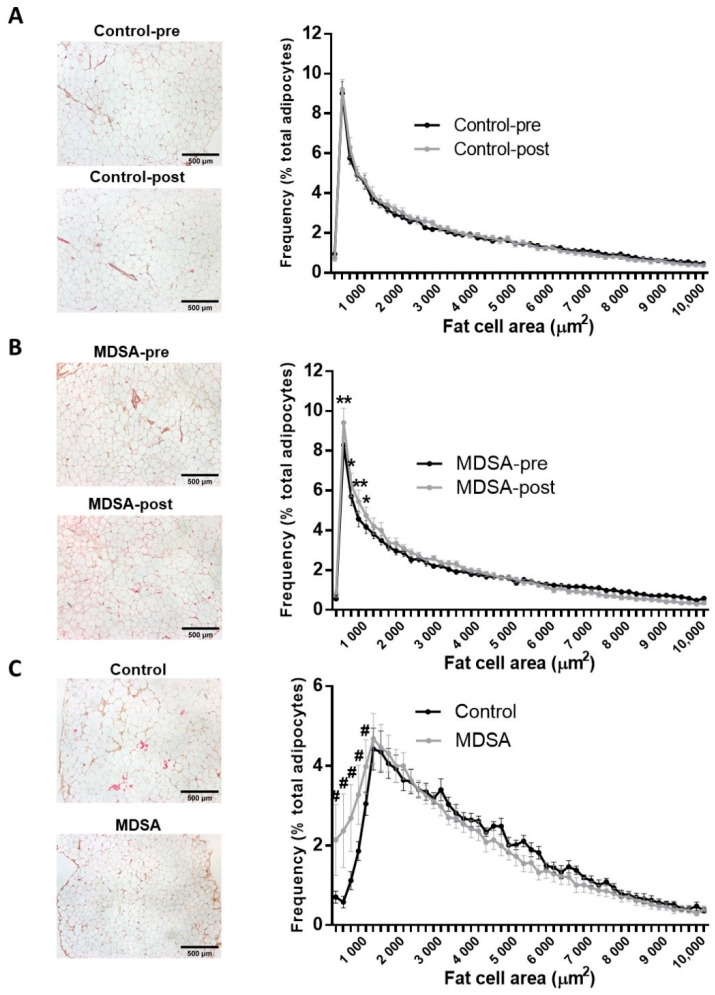
Fat cell size distribution. Comparison of frequency distribution and representative images of adipocyte cell surface area in SAT at baseline and at study completion: (**A**) Control group; (**B**) MDSA group. (**C**) Comparison of frequency distribution and representative images of fat cell areas from VAT in control and MDSA groups at the end of the study. Adipocyte areas were divided by size into bin intervals of 200 µm^2^. Data are presented as average ± SD frequencies of cells within each bin and compared by Holm–Sidak t-test for multiple comparisons. * Different from baseline; ^#^ different from control. * = *p* < 0.05, ** = *p* < 0.001, and ^#^ = *p* < 0.01 by multiple *t*-test.

**Figure 2 nutrients-14-02617-f002:**
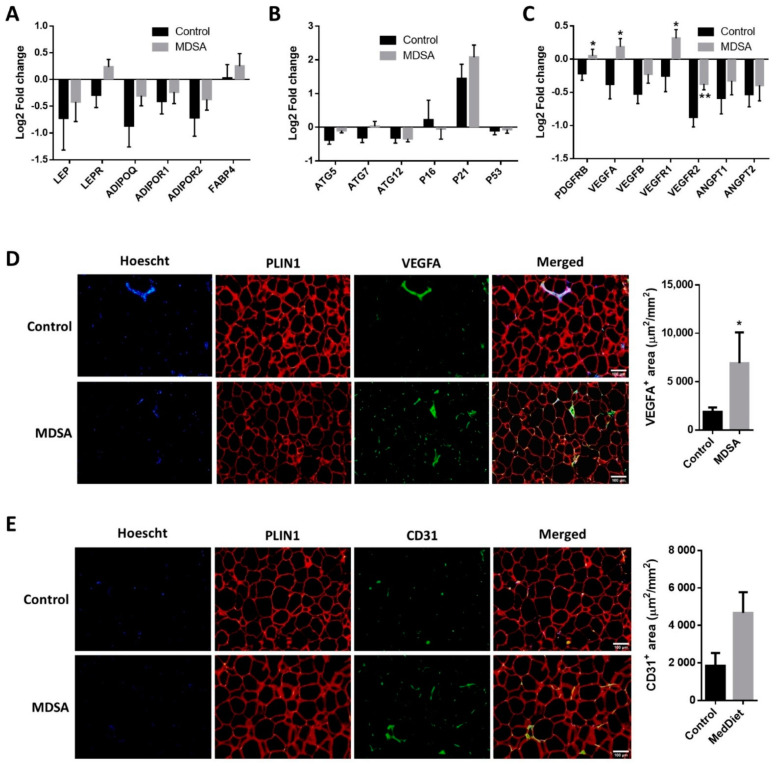
MDSA-mediated modulation of gene expression levels in SAT. Log2 fold change after control or MDSA intervention in mRNA levels of: (**A**) adipokines, (**B**) autophagy and senescence genes, and (**C**) angiogenesis genes. Data are shown as average ± SD and compared by Holm–Sidak t-test. Immunofluorescence detection and representative photomicrographs at magnification ×10 of: (**D**) VEGFA and (**E**) CD31. The counterstaining of nuclei (Hoescht) is shown in blue and PLIN1 in red. Data are presented as the average surface ± SD of positive area stained per mm^2^ and compared by Mann–Whitney U test. * = *p* < 0.05; ** = *p* < 0.01.

**Figure 3 nutrients-14-02617-f003:**
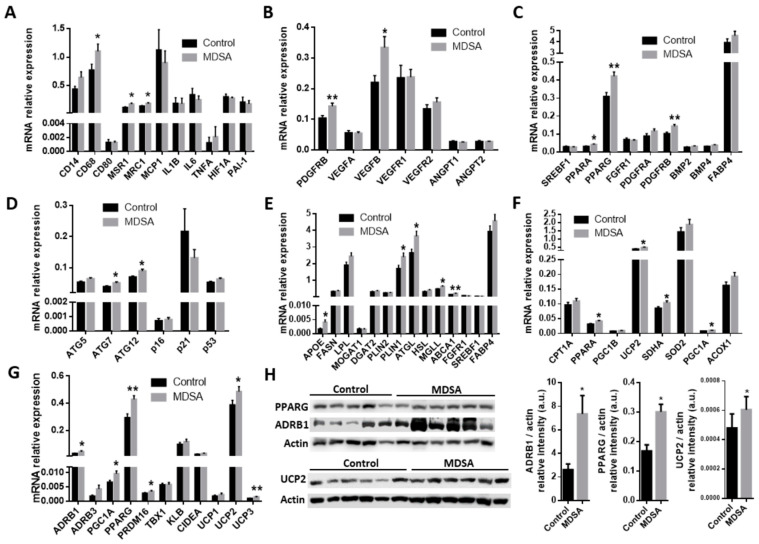
MDSA-mediated modulation of gene expression levels in VAT. Relative mRNA expression at study conclusion of genes related to: (**A**) inflammation, (**B**) angiogenesis, (**C**) adipogenesis, (**D**) autophagy and senescence, (**E**) fatty acid metabolism, (**F**) mitochondrial function and FAO, and (**G**) beiging. (**H**) Western blot analysis and optical density quantification of PPARG (57 kDa), ADRB1 (50 kDa) and UCP2 (36 kDa) contents relative to actin (40 kDa). Data are shown as average ± SD and compared to controls by Student’s *t*-test or Mann–Whitney U test for non-normally distributed data and corrected for multiple comparisons using the Holm–Sidak method. * = *p* < 0.05; ** = *p* < 0.01.

**Figure 4 nutrients-14-02617-f004:**
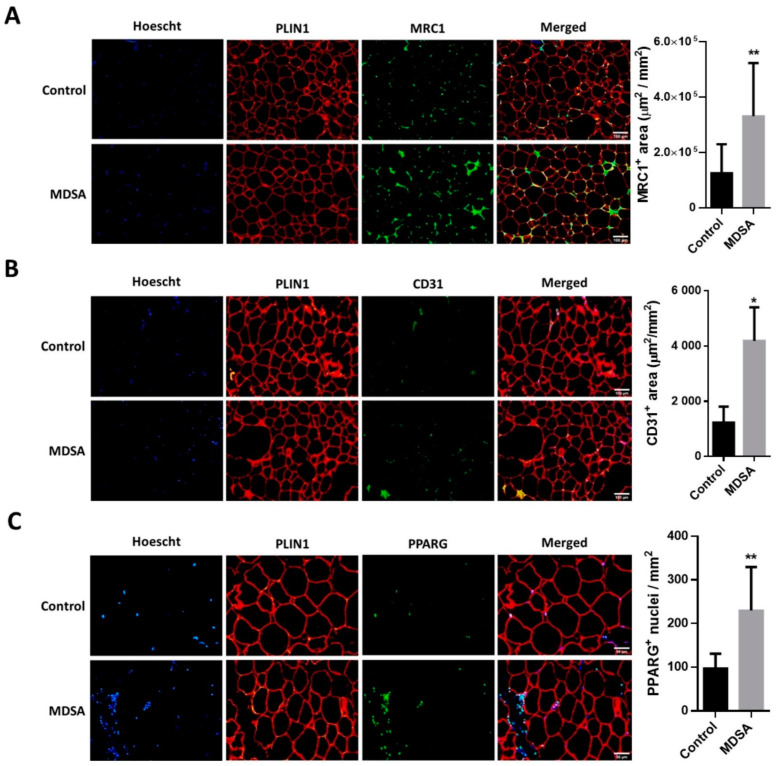
Immunofluorescence analysis in VAT and representative photomicrographs at magnification ×10 of: (**A**) MRC1+, M2-like infiltrating macrophages and (**B**) CD31 positive cells. (**C**) Immunofluorescence detection and representative photomicrographs at magnification ×20 of PPARG positive nuclei in VAT. Quantifications are presented as the average surface ± SD of positive area stained per mm^2^ and compared by Student’s *t*-test, or as the average number of positive nuclei ± SD per mm^2^ and compared by Mann–Whitney U test. The counterstaining of nuclei (Hoescht) is shown in blue and PLIN1 in red. * = *p* < 0.05; ** = *p* < 0.001.

**Figure 5 nutrients-14-02617-f005:**
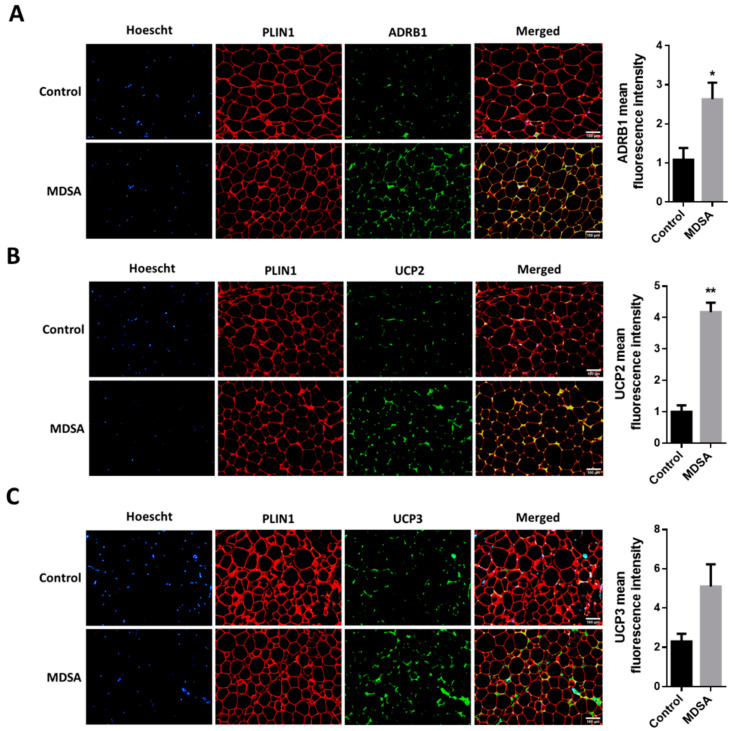
Immunofluorescence analysis in VAT and representative photomicrographs at magnification ×10 of: (**A**) ADRB1, (**B**) UCP2 and (**C**) UCP3. The counterstaining of nuclei (Hoescht) is shown in blue and PLIN1 in red. Quantifications are presented as the average fluorescence intensity ± SD and compared by Student’s *t*-test. * = *p* < 0.05, ** = *p* < 0.001.

**Table 1 nutrients-14-02617-t001:** Demographic, anthropometric, metabolic and inflammatory parameters at baseline and at the end of the study.

	MDSA Group (*n* = 19)		Control Group (*n* = 17)		*p*-Value	*p*-Value
	Baseline	3 Months	Baseline	3 Months	Time	Time * Group
Age (years)	48.9 ± 11.0	-	45.3 ± 11.8	-		
BMI (kg/m^2^)	43.8 ± 4.0	44.0 ± 3.8	45.7 ± 3.0	45.8 ± 3.1	0.276	0.769
Weight (kg)	110 ± 11.3	111 ± 11.7	110 ± 11.3	110 ± 11.9	0.243	0.643
S-BP (mmHg)	135 ± 15	132 ± 18	133 ± 16	131 ± 13	0.429	0.951
D-BP (mmHg)	86 ± 9.2	83 ± 9.5	90.3 ± 13	83 ± 8.0	0.061	0.341
FM (%)	52.8 ± 4.3	53.1 ± 3.2	55.1 ± 3.0	54.9 ± 3.0	0.705	0.274
eVAT (gr)	2351 ± 912	2261 ± 693	2396 ± 953	2262 ± 663	0.867	0.875
eVAT (cm^2^)	2492 ± 966	2540 ± 1010	2397 ± 734	2398 ± 703	0.866	0.874
Total Cholesterol (mg/dL)	187 ± 32	175 ± 24	200 ± 32	207 ± 32	0.549	0.028
HDL-c (mg/dL)	46.3 ± 8.4	46.6 ± 8.0	48.2 ± 8.3	49.7 ± 7.2	0.353	0.523
LDL-c (mg/dL)	120 ± 33	105 ± 19	125 ± 25	132 ± 30	0.364	0.012
Triglycerides (mg/dL)	124 ± 47	118 ± 30	144 ± 42.4	151.7 ± 61	0.904	0.384
FPG (mg/dL)	106 ± 14.0	106 ± 12.4	100 ± 10.6	100 ± 15.1	0.837	0.862
Insulin (uU/L)	20.1 ± 6.5	20.3 ± 6.2	24.3 ± 13.4	24.1 ± 14.1	0.998	0.894
HOMA-IR	5.3 ± 2.1	5.4 ± 2.0	6.2 ± 4.0	6.2 ± 4.1	0.985	0.884
Matsuda Index	2.3 ± 1.0	2.1 ± 1.0	2.2 ± 1.8	2.1 ± 1.4	0.354	0.9
Insulinogenic Index	2.9 ± 1.6	3.5 ± 2.1	3.3 ± 2.2	3.2 ± 1.7	0.531	0.174
Disposition Index	7.1 ± 5.1	7.4 ± 5.4	6.7 ± 5.6	6.5 ± 5.7	0.938	0.652
hsCRP (mg/dL)	0.8 ± 0.7	0.7 ± 0.6	0.8 ± 0.6	1.2 ± 1.3	0.271	0.142
GM-CSF (pg/mL)	26.6 ± 21.0	14.9 ± 7.7	15.8 ± 9.9	17.7 ± 13.7	0.089	0.018
IFNγ (pg/mL)	5.7 ± 4.5	3.2 ± 1.7	3.6 ± 3.0	4.3 ± 4.1	0.164	0.014
IL6 (pg/mL) *	2.1 ± 1.4	1.5 ± 1.0	1.2 ± 0.5	1.2 ± 0.8	0.103	0.122
TNFα (pg/mL)	4.2 ± 2.4	3.1 ± 1.2	3.6 ± 1.6	3.8 ± 1.9	0.188	0.082
IL1ß (pg/mL) *	1.2 ± 1.0	0.7 ± 0.3	0.7 ± 0.3	0.7 ± 0.4	0.049	0.024
Selectin (pg/mL)	87.2 ± 40.5	85.9 ± 45.34	86.8 ± 36.3	83.9 ± 27.2	0.391	0.749
Adiponectin (ug/mL)	19.3 ± 10.5	20.4 ± 10.1	25.1 ± 13.5	24.3 ± 15.9	0.841	0.22
sICAM-1 (ng/mL)	176 ± 110	163 ± 80.4	178 ± 91	170 ± 86.7	0.332	0.831
sVCAM-1 (ng/mL)	679 ± 162	633 ± 148	683 ± 141	666 ± 115	0.303	0.618
SAA (ug/mL)	26.9 ± 26.3	38.7 ± 42.7	62.3 ± 68.6	60.6 ± 62	0.489	0.351

Data expressed as mean ± SD. * *p* < 0.05 comparing MDSA versus control groups at baseline evaluation. BMI, Body Mass Index; S-BP, systolic blood pressure; D-BP, diastolic blood pressure; FM, fat mass; eVAT, estimated visceral adipose tissue; HDLc, High-Density Lipoprotein cholesterol; LDLc, Low-Density Lipoprotein cholesterol; FPG, Fasting Plasma Glucose; HOMA-IR, Homeostatic Model Assessment; hsCRP, high sensitive C-reactive protein; GM-CSF, Granulocyte-macrophage colony-stimulating factor; INF, Interferon; IL, Interleukin; TNF, Tumor Necrosis Factor; sICAM-1, soluble Intercellular Adhesion Molecule; sVCAM, soluble Vascular Cell Adhesion Molecule; SAA, Serum Amyloid A.

**Table 2 nutrients-14-02617-t002:** Correlation between Mediterranean Diet Adherence Screener (MEDAS) score and circulating inflammatory markers and adipose tissue gene expression levels.

	Correlation Coefficient (95%CI)	*p*-Value
VAT-ABCA1		
Unadjusted *	0.636 (0.269, 0.842)	0.002
Adjusted **	36.62 (13.07, 60.16)	0.004
VAT-UCP3		
Unadjusted *	0.579 (0.169, 0.818)	0.007
Adjusted **	1596.6 (−678.3, 3871.6)	0.160
sICAM-1 (ng/mL)		
Unadjusted *	−0.375 (−0.659, 0.0009)	0.04
Adjusted **	−0.008 (−0.024, 0.009)	0.357
VAT-PPARA		
Unadjusted *	0.350 (−0.109, 0.686)	0.119
Adjusted **	173.29 (21.41, 325.1)	0.027
VAT-PGC1A		
Unadjusted *	0.232 (−0.165, 0.564)	0.234
Adjusted **	539.9 (109.5, 970.3)	0.016
VAT-ADRB3		
Unadjusted *	0.291 (−0.103, 0.606)	0.132
Adjusted **	411.7 (60.24, 763.2)	0.023
VAT-ADRB1		
Unadjusted *	0.236 (−0.161, 0.568)	0.225
Adjusted **	69.17 (9.218, 129.14)	0.025

CI, confidence interval; VAT, visceral adipose tissue; ABCA1, ATP-binding cassette transporter A1; UCP3, mitochondrial uncoupling protein 3; sICAM-1, soluble intercellular adhesion molecule-1; PPARA, peroxisome proliferator-activated receptor alpha; ADRB1, adrenoceptor beta 1; ADRB3, adrenoceptor beta 3. * R (95%CI) from the Spearman correlation. ** Multiple linear regression Beta coefficient (95%CI) adjusted for age and BMI post-intervention.

## Data Availability

All data presented in this study are reported in this manuscript or available in Appendix A.

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
