# Peer review of "Positive Effects of a Mediterranean Diet Supplemented with Almonds on Female Adipose Tissue Biology in Severe Obesity"

_nutrients, 2022, doi:10.3390/nu14132617_

Round 1

Reviewer 1 Report

The manuscript by Osorio-Conles, et.al., entitle as "Positive effects of a Mediterranean Diet Supplemented with Almonds on Female Adipose Tissue Biology in Severe Obesity". Authors have studied Mediterranean diet supplemented with almonds for 3 months, have a positive effect on severe obesity in females, reduction in inflammation genes expression, increased thermogenesis regulating genes (UCP2). Overall, authors have found that Mediterranean diet supplemented with almonds (30g/day) have a protective effect.

Here are my comments to the authors. MAJOR

1.     The sample size is too low to conclude that almonds have some effect, total cholesterol, HDL, LDL levels are non- significant.

2.     Diet plan and composition is not provided, how the patients received almonds, raw almonds or boiled or soaked? And what components in almonds may responsible for these effects? 

3.     Authors have showed various types of genes expression and multiple images. But we must accept that always gene expression may not become protein level expression. If possible, add some more immunoblots like Fig. 3H.

4.     Did authors have checked, glucose tolerance, insulin tolerance?

5.     After 3-moths of study, there is no significant changes in bodyweight. Still the patients are obese and slight reduction in inflammation observed, how MDSA diet can help further?

6.     Did authors follow-up patients receive MDSA after 3-months of study? Found any reversible changes? 

7. Why authors didn't check any hormonal changes (estrogen/progesterone)

Author Response

  1. The sample size is too low to conclude that almonds have some effect, total cholesterol, HDL, LDL levels are non- significant.

As we acknowledge in the study limitations paragraph (lines 478-495), although the study population was homogeneous (limited to women) and groups were well matched for body composition and laboratory data, the number of subjects included is low and could have shadowed our ability to find meaningful changes. Nevertheless, after applying appropriate statistical tests, LDL and total cholesterol levels decreased in the MDSA group with respect to controls after the intervention (Table 1).

  1. Diet plan and composition is not provided, how the patients received almonds, raw almonds or boiled or soaked? And what components in almonds may responsible for these effects? 

Thank you for pointing this out. The following sentence has been modified (line118): Participants in the MDSA group received raw unpeeled almonds at no cost for the entire study (equivalent to 30 g/d).

The Diet plan and composition are now further explained in lines 119-129.

The following text has been added: The nutritional composition of almonds per 100g was: 628 kcal, total fat 56 g, saturated fat 4.9 g, carbohydrate 2.2 g, sugars 2 g, dietary fiber 9.8 g, protein 24 g, salt 0 g, vitamin E 17 mg, calcium 223 mg, phosphorus 458 mg, magnesium 232 mg, iron 3 mg. Instructions were given about how to increase MD adherence in the MDSA group in order to increase the use of olive oil for cooking and dressing, consumption of fruit, vegetables, fish and white meat instead of red or processed meat and to promote the preparation of homemade sauce with tomato, aromatic herbs, onion, garlic, and olive oil to dress vegetables, pasta, rice, or other dishes. At study inclusion and every 2 weeks, dieticians delivered individual sessions consisting of informative talks and provided written material with elaborated descriptions of typical MD foods, seasonal shopping lists, meal plans and recipes.

About 80% of the fat present in almonds is in the form of unsaturated lipids, mostly monounsaturated which may contribute to lowering total LDL cholesterol levels. As we state in the Discussion (lines 455-464), almonds are rich in α-tocopherol and polyphenols, having both well-documented activity in reducing inflammation and oxidative stress This information is now more clearly explained in the Introduction (lines 53-59) and is further discussed in the Discussion section.

The following references have been added:

  1. Bolling, B.W. Almond Polyphenols: Methods of Analysis, Contribution to Food Quality, and Health Promotion. Compr. Rev. Food Sci. Food Saf. 2017, 16, 346–368, doi:10.1111/1541-4337.12260.
  2. Garrido, I.; Urpi-Sarda, M.; Monagas, M.; Gómez-Cordovés, C.; Martín-Álvarez, P.J.; Llorach, R.; Bartolomé, B.; Andrés-Lacueva, C. Targeted analysis of conjugated and microbial-derived phenolic metabolites in human urine after consumption of an almond skin phenolic extract. J. Nutr. 2010, 140, 1799–1807, doi:10.3945/JN.110.124065.
  3. Kamil, A.; Chen, C.Y.O. Health benefits of almonds beyond cholesterol reduction. J. Agric. Food Chem. 2012, 60, 6694–6702, doi:10.1021/JF2044795.
  4. Chen, C.Y.O.; Holbrook, M.; Duess, M.A.; Dohadwala, M.M.; Hamburg, N.M.; Asztalos, B.F.; Milbury, P.E.; Blumberg, J.B.; Vita, J.A. Effect of almond consumption on vascular function in patients with coronary artery disease: a randomized, controlled, cross-over trial. Nutr. J. 2015, 14, doi:10.1186/S12937-015-0049-5.
  5. Waniek, S.; di Giuseppe, R.; Plachta-Danielzik, S.; Ratjen, I.; Jacobs, G.; Koch, M.; Borggrefe, J.; Both, M.; Müller, H.P.; Kassubek, J.; et al. Association of Vitamin E Levels with Metabolic Syndrome, and MRI-Derived Body Fat Volumes and Liver Fat Content. Nutrients 2017, 9, doi:10.3390/NU9101143.
  6. Barreca, D.; Nabavi, S.M.; Sureda, A.; Rasekhian, M.; Raciti, R.; Silva, A.S.; Annunziata, G.; Arnone, A.; Tenore, G.C.; Süntar, Ä°.; et al. Almonds (Prunus Dulcis Mill. D. A. Webb): A Source of Nutrients and Health-Promoting Compounds. Nutrients 2020, 12, 672, doi:10.3390/nu12030672.
    1. Authors have showed various types of genes expression and multiple images. But we must accept that always gene expression may not become protein level expression. If possible, add some more immunoblots like Fig. 3H.

    We agree that a discrepancy between the protein content and the abundance of its mRNA is frequently observed. Nevertheless, most relevant changes in gene expression found in our study have been further complemented by immunofluorescence (VEGFA, CD31, MRC1, PPARG, UCP3), or immunoblotting quantification, and sometimes both (PPARG, UCP2, ADRB1). Thus, we have discarded significant changes in UCP3 protein content despite its increased mRNA expression in MDSA group and its correlation with MD adherence. Sadly, some of the antibodies that we used only detect the native structure of the protein in the slides, but not the denatured form in the protein lysate.

    1. Did authors have checked, glucose tolerance, insulin tolerance?

    HOMA-IR and Matsuda, Insulinogenic and disposition indices are provided in Table 1. These results are reported in lines 365-367 and discussed in lines 402-405 and 473-477.

    1. After 3-moths of study, there is no significant changes in bodyweight. Still the patients are obese and slight reduction in inflammation observed, how MDSA diet can help further?

    As we mentioned in the Introduction, it has been shown that the benefits of MD on lipid profile and systemic inflammation occur even in the absence of significant weight loss. Thus, in clinical practice, adopting a MD can be more attainable than pursuing weight loss. Indeed, total caloric intake was estimated and adjusted to maintain body weight stable throughout the intervention (Methods, line 116).

    1. Did authors follow-up patients receive MDSA after 3-months of study? Found any reversible changes? 

      For all participants, the study ended the moment they underwent bariatric surgery after the 3-month nutrition intervention, when VAT and SAT samples were collected. Surgical and postsurgical factors may shadow the effects of the nutrition intervention from this point. On the other hand, the most significant changes were found in VAT, which will not be accessible after bariatric surgery. Anyhow, anthropometrical and circulating parameters are being collected as part of the regular follow-up of bariatric patients and will be evaluated in the future.

      1. Why authors didn't check any hormonal changes (estrogen/progesterone)

      We agree with the reviewer, it would have been interesting to explore this aspect. Unfortunately, in this exploratory study, we prioritized the quantification of inflammatory and vascular circulating parameters. We will take it into account in future studies.

Reviewer 2 Report

This manuscript is well written and describes an interesting exploratory observation related to the intake of a Mediterranean diet supplemented with almonds on female adipose tissue biology. A treat to read, my compliments! The authors, however, should exercise a little more caution in inferring health benefits from this intervention. Sound nutritional science is – among others - built on the premises that 1) foods/constituents are well defined and characterized, 2) their effect is a proven beneficial effect in the (target) population and 3) the   cause-and-effect   relationship   is   established by considering the strength, consistency, specificity, dose-response, and biological plausibility of the relationship. In a revision, I would appreciate if the authors to further define and characterize their intervention (e.g. nutritional profile of almonds) and elaborate on the limitations of their study to infer health benefits of this intervention.

Author Response

I would appreciate if the authors to further define and characterize their intervention (e.g. nutritional profile of almonds) and elaborate on the limitations of their study to infer health benefits of this intervention

Thank you for this interesting suggestion. The nutrition intervention is now more clearly explained in lines 111-129. The following text has been added:

“The nutritional composition of almonds per 100g was: 628 kcal, total fat 56 g, saturated fat 4.9 g, carbohydrate 2.2 g, sugars 2 g, dietary fiber 9.8 g, protein 24 g, salt 0 g, vitamin E 17 mg, calcium 223 mg, phosphorus 458 mg, magnesium 232 mg, iron 3 mg. Instructions were given about how to increase MD adherence in the MDSA group in order to increase the use of olive oil for cooking and dressing, consumption of fruit, vegetables, fish and white meat instead of red or processed meat and promote the preparation of homemade sauce with tomato, aromatic herbs, onion, garlic, and olive oil to dress vegetables, pasta, rice, or other dishes. At study inclusion and every 2 weeks, dieticians delivered individual sessions consisting of informative talks and provided written material with elaborated descriptions of typical MD foods, seasonal shopping lists, meal plans and recipes.”

The limitations of the study are addressed in lines 473-495.

About 80% of the fat present in almonds is in the form of unsaturated lipids, mostly monounsaturated which may contribute to lowering total LDL cholesterol levels. As we state in the Discussion (lines 456-464), almonds are rich in α-tocopherol and polyphenols, having both well-documented activities in reducing inflammation and oxidative stress This information is now more clearly explained in the Introduction (lines 53-59) and further discussed the Discussion.

The following references have been added:

  1. Bolling, B.W. Almond Polyphenols: Methods of Analysis, Contribution to Food Quality, and Health Promotion. Compr. Rev. Food Sci. Food Saf. 2017, 16, 346–368, doi:10.1111/1541-4337.12260.
  2. Garrido, I.; Urpi-Sarda, M.; Monagas, M.; Gómez-Cordovés, C.; Martín-Álvarez, P.J.; Llorach, R.; Bartolomé, B.; Andrés-Lacueva, C. Targeted analysis of conjugated and microbial-derived phenolic metabolites in human urine after consumption of an almond skin phenolic extract. J. Nutr. 2010, 140, 1799–1807, doi:10.3945/JN.110.124065.
  3. Kamil, A.; Chen, C.Y.O. Health benefits of almonds beyond cholesterol reduction. J. Agric. Food Chem. 2012, 60, 6694–6702, doi:10.1021/JF2044795.
  4. Chen, C.Y.O.; Holbrook, M.; Duess, M.A.; Dohadwala, M.M.; Hamburg, N.M.; Asztalos, B.F.; Milbury, P.E.; Blumberg, J.B.; Vita, J.A. Effect of almond consumption on vascular function in patients with coronary artery disease: a randomized, controlled, cross-over trial. Nutr. J. 2015, 14, doi:10.1186/S12937-015-0049-5.
  5. Waniek, S.; di Giuseppe, R.; Plachta-Danielzik, S.; Ratjen, I.; Jacobs, G.; Koch, M.; Borggrefe, J.; Both, M.; Müller, H.P.; Kassubek, J.; et al. Association of Vitamin E Levels with Metabolic Syndrome, and MRI-Derived Body Fat Volumes and Liver Fat Content. Nutrients 2017, 9, doi:10.3390/NU9101143.
  6. Barreca, D.; Nabavi, S.M.; Sureda, A.; Rasekhian, M.; Raciti, R.; Silva, A.S.; Annunziata, G.; Arnone, A.; Tenore, G.C.; Süntar, Ä°.; et al. Almonds (Prunus Dulcis Mill. D. A. Webb): A Source of Nutrients and Health-Promoting Compounds. Nutrients 2020, 12, 672, doi:10.3390/nu12030672.
